# Using HScore for Evaluation of Hemophagocytosis in Multisystem Inflammatory Syndrome Associated with COVID-19 in Children

**DOI:** 10.3390/biomedicines12020294

**Published:** 2024-01-27

**Authors:** Ilia S. Avrusin, Natalia N. Abramova, Konstantin E. Belozerov, Liudmila V. Bregel, Olesya S. Efremova, Alla A. Vilnits, Julia E. Konstantinova, Eugenia A. Isupova, Tatiana L. Kornishina, Vera V. Masalova, Olga V. Kalashnikova, Vyacheslav G. Chasnyk, Yuriy S. Aleksandrovich, Dmitri O. Ivanov, Mikhail M. Kostik

**Affiliations:** 1Hospital Pediatrics, Saint Petersburg State Pediatric Medical University, Saint Petersburg 194100, Russia; biancolago@bk.ru (K.E.B.); miaposta@list.ru (E.A.I.); tk.06@mail.ru (T.L.K.); masalova.vera@gmail.com (V.V.M.); koira7@yandex.ru (O.V.K.); chasnyk@gmail.com (V.G.C.); 2Intensive Care Unit Department, Saint Petersburg State Pediatric Medical University, Saint Petersburg 194100, Russia; abrnatalia@yandex.ru (N.N.A.); jalex1963@mail.ru (Y.S.A.); 3Department of Pediatrics, Irkutsk State Medical Academy of Postgraduate Education, Branch of Russian Medical Academy of Continuous Professional Education, Irkutsk 664049, Russia; loudmilabregel@yandex.ru (L.V.B.); shaguno@mail.ru (O.S.E.); 4Department of Cardiology, Irkutsk Regional Children’s Hospital, Irkutsk 664022, Russia; 5Pediatric Infectious Department, Saint Petersburg State Pediatric Medical University, Saint Petersburg 194100, Russia; vilnitz@mail.ru; 6Pediatric Research and Clinical Center for Infection Diseases, Saint Petersburg 197022, Russia; yulia.konstantinova23@mail.ru; 7Neonatology Department, Saint Petersburg State Pediatric Medical University, Saint Petersburg 194100, Russia

**Keywords:** children, COVID-19, ICU, hemophagocytosis, MIS-C

## Abstract

Hemophagocytic syndrome is a key point in the pathogenesis of severe forms of multisystem inflammatory syndrome associated with COVID-19 in children (MIS-C). The factors associated with hemophagocytosis in patients with MIS-C were assessed in the present study of 94 boys and 64 girls ranging in age from 4 months to 17 years, each of whose HScore was calculated. In accordance with a previous analysis, patients with HScore ≤ 91 (*n* = 79) and HScore > 91 (*n* = 79) were compared. Patients with HScore > 91 had a higher frequency of symptoms such as cervical lymphadenopathy, dry cracked lips, bright mucous, erythema/swelling of hands and feet, peeling of fingers, edematous syndrome, hepatomegaly, splenomegaly, and hypotension/shock. They also had a higher erythrocyte sedimentation rate (ESR), C-reactive protein (CRP) and D-dimer levels, and a tendency to anemia, thrombocytopenia, and hypofibrinogenemia. They more often needed acetylsalicylic acid and biological treatment and were admitted to ICU in 70.9% of cases. Conclusion: The following signs of severe MIS-C were associated with HScore > 91: myocardial involvement, pericarditis, hypotension/shock, and ICU admission.

## 1. Introduction

Multisystem inflammatory syndrome associated with COVID-19 in children (MIS-C) is a rare, serious disease (the frequency is 1:3000–4000 children who have had COVID-19 [1,2]). It affects the heart, lungs, kidneys, skin, eyes, nervous system, and organs of the digestive system, often simultaneously [3,4,5]. In most cases, MIS-C develops approximately 3–4 weeks after COVID-19 [5,6,7,8].

The pathophysiological mechanism of this condition is still not clear. It could be due to immune dysregulation leading to anomalous immune response to the virus, itself leading to massive production of cytokines such as interleukin-1β (IL)-1β, IL-6, IL-8, IL-10, and IL-18 [9,10].

The diagnosis and differential diagnosis of MIS-C is often complex because of its numerous clinical manifestations and considerable similarity with Kawasaki disease (KD).

In some cases, MIS-C patients closely resemble patients with KD due to similar clinical symptoms and even meet the criteria of the American Heart Association (AHA) for KD, but these diseases are different [4,5,7,8,11,12].

MIS-C is more often observed in older children than KD. The involvement of the heart (myocarditis and pericarditis), often leading to systolic myocardial dysfunction and shock [13,14,15,16] along with gastrointestinal involvement (diarrhea, abdominal pain, and vomiting) are the main clinical features of severe MIS-C, as well as inflammatory markers, thrombocytopenia, hypoalbuminemia, and high levels of D-dimer [4,5,6,7,8,13,17,18].

MIS-C often requires hospitalization in the intensive care unit (ICU) (about 50%, according to several studies [7,12,13,15,16,19,20]).

Hemophagocytic syndrome was proven to be one of the most important factors determining the severity of the course of MIS-C [13,20,21]. It can also be considered a complication of Kawasaki disease [22]. The signs of secondary hemophagocytic lymphohistiocytosis (HLH) are significantly more frequent in MIS-C than in KD (18.4% vs. 1.2%, respectively), according the literature [23].

The purpose of this study was to determine the features of hemophagocytosis in patients with multisystem inflammatory syndrome associated with COVID-19 in children. 

## 2. Materials and Methods

### 2.1. Patients

The data from case histories (*n* = 166) of children with MIS-C associated with COVID-19 diagnosed between 2020 and 2022 were included in the present retrospective multi-central study. The diagnosis of MIS-C was made according to the criteria of the CDC and the WHO [24,25]. 

Previous COVID-19 infection was confirmed with a positive result from a PCR test with reverse transcription (13%), Ig M (40.3%), or Ig G (97.4%) antibodies to SARS-CoV-2, or close contact (family, school, or kindergarten) with a person with confirmed COVID-19 (65.6%).

We excluded eight patients with missing data that did not allow confirmation or exclusion of hemophagocytosis and cases with an obvious microbial cause of multisystem inflammation (e.g., bacterial sepsis and staphylococcal/streptococcal toxic shock syndrome).

### 2.2. Study Design

We included all the related clinical data and the most significant laboratory test abnormalities obtained close to the time of hospital admission.

For diagnosis of hemophagocytic syndrome, we used the following diagnostic sets: (a) HScore calculation (Table 1), (b) HLH-2004 criteria, and (c) 2005 and 2016 EULAR/ACR/PRINTO criteria for macrophage activation syndrome in children with systemic juvenile idiopathic arthritis [26,27,28,29].

According to data from the previous analysis, patients with HScore ≤ 91 (n = 79) and HScore > 91 (n = 79) were compared because this cut-off is associated with the severe life-threatening course of MIS-C [20].

### 2.3. Statistics

The STATISTICA software package, version 10.0 (StatSoft Inc., Tulsa, OK, USA), was used for the data analysis. The median value (25th and 75th percentiles) was used to assess quantitative indicators. The Pearson criterion χ2 was used to compare qualitative indicators. Quantitative indicators were compared based on the Mann–Whitney criterion. Sensitivity and specificity analysis was used to evaluate the possibility of patient grouping based on binary characteristics. For quantitative variables, cut-off values were calculated using AUC-ROC analysis (AUC) with a 95% confidence interval (CI). The odds ratio (OR) calculation, independent of the time of event development, was performed using 2 × 2 tables. Sensitivity (Se) and specificity (Sp) were assessed for each parameter. Binary logistic regression was used to specify independent predictors, including quantitative and qualitative indicators associated with the dependent variable in the analysis. Differences or relatedness were considered statistically significant at *p* < 0.05.

## 3. Results

### 3.1. General Characteristics of MIS-C Patients

One hundred and fifty eight (158) patients (94 boys and 64 girls) were included in the analysis. Boys suffered more frequently (59.6%) than girls. The patients were of different ages (from 4 months to 17 years; the median age was 8 years and 2 months). 

The following symptoms of MIS-C were seen most often: fever (100%), conjunctivitis (84.8%), rash (78.9%), gastrointestinal disorders (77.2%), lymphadenopathy (66.9%), mucosal hyperemia (64%), hepatomegaly (64.4%), erythema/edema of hands and feet (62.4%), sore throat (56.3%), edematous syndrome (50.5%), breathing disorders (49.4%), lips blushing and parching (49.3%), neurological disturbance (47.8%), hypotension/shock (43.8%), and splenomegaly (40.7%).

Most frequently, the following laboratory abnormalities were observed: signs of inflammation such as erythrocyte sedimentation rate (ESR) (n = 150, 90.9%), C-reactive protein (CRP) (n = 153/157, 97.5%), ferritin (n = 69/90, 76.7%), increased liver enzymes (alanine aminotransferase (ALT) (n = 85/164, 52.8%), aspartate aminotransferase (AST) (n = 101/148, 68.2%), lactate dehydrogenase (LDH) (n = 56/94, 59.6%), D-dimer (n = 120/125, 96.0%), hypoalbuminemia (n = 115/182, 81.0%), and hypoproteinemia (n = 102/126, 81.0%). Signs of hemophagocytosis (thrombocytopenia (n = 79, 47.6%), hyperferritinemia, increased liver enzymes, increased LDH and D-dimer, hypoalbuminemia, and hypoproteinemia) were often found.

Echo-signs of heart disorders were also quite common, namely myocardial damage (30.6%), pericardial effusion (28.8%), and coronary artery (CA) dilation/aneurysms (15.8%).

Standard treatment in MIS-C patients was administered as follows: glucocorticosteroids (81.5%), acetylsalicylic acid (57.1%), intravenous immunoglobulin (44.7%, due to restricted access), and tocilizumab (4.9%). Supportive care was also administered via albumin, plasma, erythrocyte mass, and infusion and inotropic therapy. Patients stayed in the hospital for 18 days on average. No fatal cases were observed. All patients returned to full health.

### 3.2. Signs of Hemophagocytosis in MIS-C Patients

The median age of the patients was 104 and 105 months, respectively. Male patients predominated in both groups (62% and 57%, respectively). All characteristics of both groups of patients are presented in Table 2.

Patients with HScore > 91 had a higher frequency of such symptoms as cervical lymphadenopathy (80.6% vs. 54.1%, OR = 3.5 (95%CI: 1.7; 2.4), *p* = 0.0007), dry cracked lips (63% vs. 34.3%, OR = 3.3 (95%CI: 1.6; 6.5), *p* = 0.0007), bright mucous membranes (72.7% vs. 52.9%, OR = 2.4 (95%CI: 1.1; 5.3), *p* = 0.03), erythema or swelling of hands and feet (70.3% vs. 52.2%, OR = 2.2 (95%CI: 1.1; 4.3), *p* = 0.03), peeling of fingers (53.5% vs. 17.2%, OR = 5.6 (95%CI: 2.5; 12.3), *p* = 0.00001) edematous syndrome (66.7% vs. 34.7%, OR = 3.8 (95%CI: 1.7; 8.5), *p* = 0.001), hepatomegaly (84.2% 43.1%, OR = 7.1 (95%CI: 3.2; 15.5), *p* < 0.0000001), splenomegaly (54.7% vs. 43.1%, OR = 3.7 (95%CI: 1.8; 7.6), *p* = 0.0003), and hypotension/shock (63.3% vs. 25.3%, OR = 5.7 (95%CI: 2.9; 11.2), *p* = 0.000002).

Among laboratory parameters, differences in the levels of inflammatory markers were found, so ESR (47 mm/h vs. 34 mm/h, *p* = 0.0001) and CRP (175.5 mg/L vs. 125.8 mg/L, *p* = 0.01) were more significantly elevated in patients with Hscore > 91.

There was also, on average, lower levels of red blood cells (3.6 × 10^12^/L vs. 4.0 × 10^12^/L, *p* = 0.000005) and hemoglobin (98 g/L vs. 112 g/L, *p* < 0.0000001), and a tendency to thrombocytopenia (110 × 10^9^/L vs. 192 × 10^9^/L, *p* = 0.0002) in 63.3% of patients. Transaminase levels were also higher in patients with HScore > 91: ALT (47.5 U/L vs. 36.5 U/L, *p* = 0.015) and AST (55.0 U/L vs. 44.1 U/L, *p* = 0.029). In patients with HScore > 91, there was a more significant increase in the level of D-dimer (2135 ng/mL vs. 1079 ng/mL, *p* = 0.0003); conversely, the level of fibrinogen was slightly lower (3.1 g/L vs. 5.6 g/L, *p* = 0.000002).

Heart damage was noted significantly more often in patients with HScore > 91 (70.9% vs. 32.9%, OR = 4.7 (95%CI: 2.4; 9.2), *p* = 0.000002). According to echocardiographic data, signs of myocardial damage (45.5% vs. 15.6%, OR = 4.5 (95%CI: 2.1; 9.7), *p* = 0.00006) and pericarditis (45.5% vs. 14.3%, OR = 5.0 (95%CI: 2.3; 10.9), *p* = 0.00002) were seen significantly more often in this group of patients, and dilation/aneurysms of the coronary arteries were equally common (16%).

It is worth noting that in the treatment of patients with HScore > 91, intravenous immunoglobulin was significantly more frequently prescribed (66.2% vs. 24%, OR = 6.2 (95%CI: 3.1; 12.6), *p* < 0.0000001), as well as therapy with acetylsalicylic acid (65.7% vs. 47.1%, OR = 2.2 (95% CI; 1.1; 4.3), *p* = 0.027) and biologics (9.1% vs. 1.6%, OR = 6.3 (95%CI: 0.7; 55.7), *p* = 0.061).

Of the patients with HScore > 91, 70.9% needed ICU admission, which is twice the rate seen in patients with HScore ≤ 91 (32.3%, OR = 4.7 (95%CI: 2.4; 9.2), *p* = 0.000002). The patients with HScore > 91 stayed in hospital for a much longer time, so the median duration of hospitalization was 23 days and 14 days, respectively (*p* = 0.0000001).

Patients with HScore > 91 more often required ICU admission, the use of intravenous immunoglobulins, and biologic drugs. The clinical and laboratory signs associated with HScore > 91 are outlined in Table 3.

## 4. Discussion

In our study, we calculated the threshold value of the HScore index associated with more severe forms of MIS-C. Criteria of severity of MIS-C such as cardiac involvement, including myocardial and pericardial damage and hypotension/shock, were associated with intense hemophagocytosis (HScore > 91). Patients with intensive hemophagocytosis more often required the use of intravenous immunoglobulins and biological drugs.

The incidence of hemophagocytic syndrome in MIS-C is 18–76%, according to various studies [5,13,21,23,30,31]. In our study, signs of hemophagocytic syndrome were recorded in 35.4% of patients and predominantly in patients with a severe course of disease which required ICU admission (48.8%). A similar pattern was described in other studies [30,31]. The frequency of cytokine storm syndrome (CSS) in MIS-C and active COVID-19 depended on the scoring systems applied to identify macrophage activation syndrome (MAS) and hemophagocytic lymphohistiocytosis (HLH). Its frequency in patients with MIS-C was as high as 76% in an American study [30]. We also used some of these scoring systems. For example, in our study, 5.7% of patients met the HLH-2004 criteria, and 11.5% fulfilled the criteria for MAS (Ravelli, 2016). In the study mentioned above, we can see quite comparable statistics: 2% of patients met the HLH-2004 criteria and 24% the 2016 sJIA/MAS criteria.

Every fifth MIS-C patient (21%) fulfilled the 2016 MAS criteria, which is almost the same as in our study (24%), as well as exhibiting comparable frequency of swelling or erythema of hands and feet (75.5 vs. 49.8% and 70.3% vs. 52.2% in our study) [31]. Among laboratory changes, there were low platelet counts (140 vs. 188 10^3^/μL, and 110 × 10^3^/μL vs. 192 × 10^3^/μL) and albumin levels (3.1 vs. 3.4 g/dL, and 2.8 vs. 3.0 g/dL) compared to our results, respectively. Children with MAS had significantly higher median concentrations of C-reactive protein, ferritin, D-dimer, and triglycerides [31].

Regarding approaches to therapy, in our study, patients with hemophagocytosis (HScore > 91) were significantly more often treated with intravenous immunoglobulin (66.2% vs. 24%, *p* < 0.0000001), acetylsalicylic acid (65.7% vs. 47.1%, *p* = 0.027), and biologics (9.1% vs. 1.6%, *p* = 0.061). Also, patients with HScore > 91 required hospitalization in the ICU twice as often as patients with HScore ≤ 91 (70.9% vs. 32.3%, *p* = 0.000002).

A similar pattern was observed in the research of Buda et al.: MIS-C patients fulfilling the MAS criteria were treated more often with IVIG (94.8 vs. 87.6%) and steroids (87.3 vs. 62.7%) than children without MAS, and they also needed ICU admission two times as often. The median hospitalization time was also significantly longer in patients with hemophagocytosis in our study (23 days vs. 14 days, *p* < 0.0000001) and also according to the data from Buda et al. (20 days vs. 12 days) [31].

The dissimilarities in treatment are related to different national recommendations. Our patients received glucocorticosteroids as the first-line treatment [32]; meanwhile, ACR guidelines [33] and British national consensus [34] recommend commencing treatment with IVIG, as in the treatment of Kawasaki disease.

The hemophagocytic syndrome associated with MIS-C can be attributed to MAS rather than to primary HLH. Attention is drawn to the more pronounced systemic inflammation in patients with MIS-C compared with patients with primary HLH in whom the increase in CRP is usually not as significant [35,36]. This is partly explained by the different cytokine profiles. Interferon-gamma is the leading cytokine in primary HLH, whereas IL-1, 6, and 17 lead in MIS-C [9,10,37,38]. MIS-C occupies an intermediate position between the secondary, virus-associated HLH to which it formally belongs and MAS in sJIA, which it more closely resembles. Options for the treatment of MAS in sJIA have shown the greatest effectiveness in MIS-C.

Rare heterozygous missense variants in genes responsible for primary hemophagocytic lymphohistiocytosis (*LYST*, *STXBP2*, *PRF1*, *UNC13D*, *AP3B1*, and *DOCK8*) were found in patients with MIS-C [39].

The phenomenon of hemophagocytosis/macrophage activation syndrome is a fairly frequent and very serious manifestation of MIS-C, which has been confirmed by several studies [23,30,31].

Unfortunately, at the moment there are no specific criteria for evaluating hemophagocytosis in MIS-C. Many authors use previously developed criteria for other diseases, but it is not entirely known how accurate this is [27,28,29].

It is necessary to develop and validate an individual set of criteria for assessing hemophagocytosis in patients with MIS-C [30,31]. Future assessments are being planned to find the discriminative or diagnostic criteria of MAS validated in patients with MIS-C.

However, we can say with confidence that HScore works quite well in the assessment of hemophagocytosis in patients with MIS-C.

### Study Limitations

The limitations of our study are related to the retrospective study design, missing data, relatively small sample size, different study time intervals, method of selection of clinical and laboratory parameters and different SARS-CoV-2 variants affecting patients during the pandemic, and the impossibility of having laboratory data from the same time point. We fully understand that these limitations can affect the study results.

## 5. Conclusions

Hemophagocytic syndrome was diagnosed in 35.4% of the children studied, and in 48.8% of patients with a severe course. HScore > 91 was associated with myocardial damage, pericarditis, hypotension/shock, and ICU admission. HScore is a simple tool that can also be used to assess the severity of MIS-C and for dynamic monitoring. Using this concept, we can suggest the following management scheme for patients with MIS-C (Figure 1). Nevertheless, it is necessary to develop and validate specific criteria for the assessment of hemophagocytosis in the course of MIS-C.

## Figures and Tables

**Figure 1 biomedicines-12-00294-f001:**
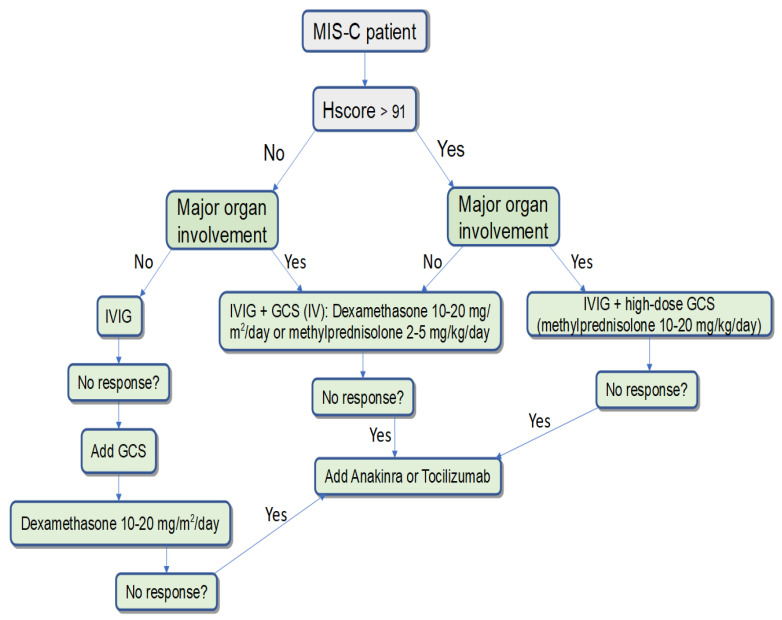
The algorithm for the management of patients with MIS-C, depending on the presence of signs of hemophagocytosis and severity of the disease.

**Table 1 biomedicines-12-00294-t001:** The HScore [26].

Parameter	No. of Points (Criteria for Scoring)
Known underlying immunosuppression *	0 (no) or 18 (yes)
Temperature (°C)	0 (<38.4), 33 (38.4–39.4), or 49 (>39.4)
Organomegaly	0 (no), 23 (hepatomegaly or splenomegaly), or 38 (hepatomegaly and splenomegaly)
No. of cytopenias **	0 (1 lineage), 24 (2 lineages), or 34 (3 lineages)
Ferritin (ng/mL)	0 (<2000), 35 (2000–6000), or 50 (>6000)
Triglyceride (mmol/L)	0 (<1.5), 44 (1.5–4), or 64 (>4)
Fibrinogen (g/L)	0 (>2.5) or 30 (≤2.5)
Aspartate aminotransferase (IU/L)	0 (<30) or 19 (>30)
Hemophagocytosis features on bone marrow aspirate	0 (no) or 35 (yes)

* Human immunodeficiency virus-positive or receiving long-term immunosuppressive therapy (i.e., glucocorticoids, cyclosporine, or azathioprine). ** Defined as a hemoglobin level of ≤9.2 g/dL and/or a leukocyte count of ≤5000/mm^3^ and/or a platelet count of ≤110,000/mm^3^.

**Table 2 biomedicines-12-00294-t002:** Comparison of characteristics of MIS-C patients with HScore > 91 and patients with HScore ≤ 91.

Parameter	Hscore ≤ 91 (*n* = 79)	Hscore> 91 (*n* = 79)	*p*-Value *
Demographics			
Age, months, Me (25%; 75%)	104 (68; 140)	105 (53; 133)	0.710
Gender, male, *n* (%)	49 (62)	45 (57)	0.517
Gender, female, *n* (%)	30 (38)	34 (43)	
COVID-19 identification			
PCR, *n* (%)	14/78 (18)	7/77 (9.1)	0.107
IgM, *n* (%)	28/62 (45.2)	23/60 (38.3)	0.445
IgG, *n* (%)	72/73 (98.6)	70/72 (97.2)	0.552
Family contact, *n* (%)	23/35 (65.7)	28/42 (66.7)	0.930
Clinical signs			
Fever, t °C, Me (25%; 75%)	39.5 (39; 40)	39.8 (39.5; 40)	0.288
GI symptoms, *n* (%)	56/75 (74.7)	66/79 (83.5)	0.175
Neurological symptoms, *n* (%)	31/75 (41.3)	40/78 (51.3)	0.217
Sore throat, *n* (%)	41/74 (55.4)	45/76 (59.2)	0.638
Rash, *n* (%)	58/71 (81.7)	58/77 (75.3)	0.347
Conjunctivitis, *n* (%)	59/70 (84.3)	65/74 (87.8)	0.538
Dry cracked lips, *n* (%)	23/67 (34.3)	46/73 (63)	**0.0007**
Bright mucous, *n* (%)	27/51 (52.9)	40/55 (72.7)	**0.035**
Respiratory signs, *n* (%)	38/76 (50)	38/78 (48.7)	0.874
Cervical lymphadenopathy, *n* (%)	40/74 (54.1)	58/72 (80.6)	**0.0007**
Hands/feet erythema/edema, *n* (%)	35/67 (52.2)	52/74 (70.3)	**0.028**
Peeling of fingers, *n* (%)	11/64 (17.2)	38/71 (53.5)	**0.00001**
Edematous syndrome, %	17/49 (34.7)	36/54 (66.7)	**0.001**
Hepatomegaly, *n* (%)	28/65 (43.1)	64/76 (84.2)	**0.000001**
Splenomegaly, *n* (%)	28/65 (43.1)	41/75 (54.7)	**0.0003**
Arthritis/arthralgia, *n* (%)	8/65 (12.3)	13/78 (16.7)	0.463
Shock/hypotension, *n* (%)	20/79 (25.3)	50/79 (63.3)	**0.000002**
Heart damage signs, *n* (%)	26/79 (32.9)	56/79 (70.9)	**0.000002**
Duration of fever, days, Me (25%; 75%)	8 (7; 12)	10 (7; 13)	0.199
KD criteria fulfillment			
Complete, *n* (%)	21/79 (26.6)	46/79 (58.2)	0.00005
Incomplete, *n* (%)	19/57 (33.3)	18/33 (54.6)	0.049
Laboratory			
Red blood cells, 10^12^/L, Me (25%; 75%)	4.0 (3.7; 4.5)	3.6 (3.3; 3.9)	**0.000005**
Hemoglobin, g/dL, Me (25%; 75%)	112 (100; 119)	98 (88; 108)	**0.000001**
White blood cells, 10^9^/L, Me (25%; 75%)	16.0 (12.8; 22.3)	16.5 (12.0; 21.5)	0.846
Platelets, 10^9^/L, Me (25%; 75%)	192 (149; 530)	110 (77; 447)	**0.0002**
Thrombocytopenia, *n* (%)	29/79 (36.7)	50/79 (63.3)	**0.0008**
ESR, mm/h, Me (25%; 75%)	34 (25; 45)	47 (34; 55)	**0.0001**
C-reactive protein, mg/dl, Me (25%; 75%)	125.8 (27.0; 209.0)	175.5 (92.5; 259.9)	**0.013**
Ferritin, μg/L, Me (25%; 75%)	282.2 (158.2; 476.3)	265.0 (78.2; 530.4)	0.629
Increased ferritin, *n* (%)	30/35 (85.7)	36/50 (72)	0.135
ALT, IU/L, Me (25%; 75%)	36.5 (23.1; 60.0)	47.5 (28.1; 83.4)	**0.015**
Increased ALT, *n* (%)	35/79 (44.3)	47/79 (59.5)	0.056
AST, IU/L, Me (25%; 75%)	44.1 (33.1; 68.2)	55.0 (37.0; 101.0)	**0.029**
Increased AST, *n* (%)	43/72 (59.7)	55/79 (78.6)	**0.015**
Bilirubin, mmol/L, Me (25%; 75%)	13.3 (7.4; 18.0)	8.3 (5.1; 14.6)	0.103
Serum protein, g/L, Me (25%; 75%)	58.0 (52.3; 63.9)	54.0 (46.0; 60.9)	**0.006**
Albumin, g/L, Me (25%; 75%)	30.4 (26.1; 34.0)	28.5 (25.0; 33.3)	0.135
Triglycerides, mmol/L, Me (25%; 75%)	1.3 (1.1; 1.5)	2.5 (1.9; 3.4)	**0.0002**
Increased triglycerides, *n* (%)	0/7 (0)	41/46 (89.1)	**0.000001**
Creatinin, mmol/L, Me (25%; 75%)	57.0 (46.6; 70.0)	52.4 (42.0; 71.9)	0.350
LDH, IU/L, Me (25%; 75%)	564.0 (279.0; 747.0)	431.5 (263.5; 637.5)	0.081
Fibrinogen, g/L, Me (25%; 75%)	5.6 (3.7; 7.9)	3.1 (1.7; 5.2)	**0.000002**
D-dimer, ng/mL, Me (25%; 75%)	1079 (569; 2345)	2135 (954; 3930)	**0.0003**
Troponin, pg/mL, Me (25%; 75%)	2.0 (0.0; 5.0)	8.0 (2.4; 100.0)	**0.031**
HLH-2004, *n* (%)	0/79 (0)	9/78 (11.5)	**0.002**
MAS 2005, *n* (%)	11/79 (13.9)	45/79 (57)	**0.000001**
MAS 2016, *n* (%)	3/79 (3.8)	15/78 (19.2)	**0.002**
Echo findings			
CA dilatation/aneurism, *n* (%)	12/77 (15.6)	12/75 (16)	0.944
Myocardial involvement, *n* (%)	12/77 (15.6)	35/77 (45.5)	**0.00006**
Pericarditis, *n* (%)	11/77 (14.3)	35/77 (45.5)	**0.00002**
Treatment and outcomes			
IVIG treatment, *n* (%)	18/75 (24)	51/77 (66.2)	**0.000001**
IVIG 2nd dose, *n* (%)	2/30 (6.7)	5/41 (12.2)	0.440
Acetylsalicylic acid, *n* (%)	33/70 (47.1)	46/70 (65.7)	**0.027**
Glucocorticosteroids, *n* (%)	62/76 (81.6)	65/79 (82.3)	0.910
Biologics, *n* (%)	1/64 (1.6)	5/55 (9.1)	0.061
ICU admission, *n* (%)	26/79 (32.3)	56/79 (70.9)	**0.000002**
Stay in hospital, in days, Me (25%; 75%)	14 (10; 17)	23 (17; 30)	**0.000001**

Abbreviations: Me—median; PCR—polymerase chain reaction; Ig—immunoglobulin; GI—gastrointestinal; KD—Kawasaki disease; ESR—erythrocyte sedimentation rate; LDH—lactate dehydrogenase; ALT—alanine aminotransferase; AST—aspartate aminotransferase; LDH—lactate dehydrogenase; HLH—hemophagocytic lymphohistiocytosis; MAS—macrophage activation syndrome; CA—coronary artery; IVIG—intravenous immunoglobulin; ICU—intensive care unit; *—comparison between those admitted and not admitted in the ICU.

**Table 3 biomedicines-12-00294-t003:** Parameters associated with hemophagocytosis (HScore > 91) in MIS-C.

Parameter	Se	Sp	OR (95%CI)	*p*
Bright oral mucosa	72.7	47.1	2.4 (1.1; 5.3)	0.035
Dry cracked lips	63.0	65.7	3.3 (1.6; 6.5)	0.0007
Cervical lymphadenopathy	80.6	45.9	3.5 (1.7; 2.4)	0.0007
Hands and feet edema/erythema	70.3	47.8	2.2 (1.1; 4.3)	0.028
Edematous syndrome	66.7	65.3	3.8 (1.7; 8.5)	0.001
Hepatomegaly	84.2	56.9	7.1 (3.2; 15.5)	0.000001
Splenomegaly	43.1	45.3	3.7 (1.8; 7.6)	0.0003
Thrombocytopenia	63.3	63.3	3.0 (1.6; 5.7)	0.0008
AST elevation	78.6	40.3	2.5 (1.2; 5.2)	0.015
Myocardial involvement	45.5	84.4	4.5 (2.1; 9.7)	0.00005
Pericardial effusion	45.5	85.7	5.0 (2.3; 10.9)	0.00002
Hypotension/shock	63.3	74.7	5.7 (2.9; 11.2)	0.000002
Any heart involvement	70.9	67.0	4.7 (2.4; 9.2)	0.000002
ICU admission	70.9	67.0	4.7 (2.4; 9.2)	0.000002
IVIG therapy	66.2	76.0	6.2 (3.1; 12.6)	0.000001
Biological therapy	9.1	98.4	6.3 (0.7; 55.7)	0.061

Abbreviations: Se—sensitivity; Sp—specificity; AST—aspartate aminotransferase; IVIG—intravenous immune globulin; ICU—intensive care unit.

## Data Availability

The datasets generated during and/or analyzed during the current study are available from the corresponding author upon reasonable request.

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
