# Peer review of "Using HScore for Evaluation of Hemophagocytosis in Multisystem Inflammatory Syndrome Associated with COVID-19 in Children"

_biomedicines, 2024, doi:10.3390/biomedicines12020294_

Round 1

Reviewer 1 Report

Comments and Suggestions for Authors

The manuscript represents an analysis of possible relationship of MS-C and possible hematophagosome. The article is essentially based on the clinical follow up of the patients no preclinical or revelent clinical data was given. When, How and how long were the symptoms are questios partially asdressed. There is a relevant issu conceerning pediatric Sars Co-2 infection and should be better analyzed. One of them is the titer of antibody against the antigens and circulating viral antigens. Another issue refers to the high inflammatory response and the therapy used. There is no clear information 

Author Response

Dear Reviewer! Thank you so much for your positive evaluations of our manuscript. Our answers (A) on your queries (Q) are below and highlighted by color in the manuscript.

Reviewer 1:

Q1. The manuscript represents an analysis of possible relationship of MS-C and possible hematophagosome. The article is essentially based on the clinical follow up of the patients no preclinical or revelent clinical data was given. When, How and how long were the symptoms are questios partially asdressed. There is a relevant issu conceerning pediatric Sars Co-2 infection and should be better analyzed.

A1. The main idea of our manuscript was to show the possibility to assess macrophage activation syndrome and related symptoms with quantitative indicator HScore. This indicator is easy to calculate and can help in the dynamic monitoring of the patient’s condition.

According to your questions: We included the clinical signs of the disease at the moment of the hospital admission and chose the most impressive laboratory tests closer to the admission date. Some missing data and manner of choosing the lab data are factors limiting the study results.

We added the information about time-point assessments and add more information in the Limitations section.

Q2. One of them is the titer of antibody against the antigens and circulating viral antigens.

A2. Dear Reviewer! We used the antibodies data only for confirmation of the etiology of MIS-C. Unfortunately, due to multi-central retrospective study analysis it was impossible to calculate the median titer of the antibodies between the centers. Also, we had no possibility to collect the samples and measured circulating antigen. It was out of this study aim.

Q3.Another issue refers to the high inflammatory response and the therapy used. There is no clear information.

A3. The main inflammatory parameters and treatment are in table 2.

Dear Reviewer!

I hope the manuscript became better after your suggestions and recommendations.

On behalf of the Authors

Mikhail Kostik, MD, PhD, Professor

Reviewer 2 Report

Comments and Suggestions for Authors

The Authors present an interesting paper: " Using HScore for evaluation of hemophagocytosis in multisystem inflammatory syndrome associated with COVID-19 in children" that could be useful for clinical approach.

Abstract: the acronym ESR and CRP as first citation should be reported as full name (even if reported later in the footnote of table 2)

Introduction as well as Material and Methods are fine

Results: In the table 2 it  the significant p-Value (when present) should be highlighted for an immediate reading

Discussion is fine

Conclusions: should report as short flow-chart the recommendations to be adopted based on their study. What procedure in term of treatment and laboratory tests are considered indispensable for the best and useful approach

References: are fine

Author Response

Reviewer 2:

Q1. The Authors present an interesting paper: " Using HScore for evaluation of hemophagocytosis in multisystem inflammatory syndrome associated with COVID-19 in children" that could be useful for clinical approach.

Abstract: the acronym ESR and CRP as first citation should be reported as full name (even if reported later in the footnote of table 2)

Introduction as well as Material and Methods are fine

Results: In the table 2 it  the significant p-Value (when present) should be highlighted for an immediate reading

Discussion is fine

Conclusions: should report as short flow-chart the recommendations to be adopted based on their study. What procedure in term of treatment and laboratory tests are considered indispensable for the best and useful approach

References: are fine

A1: Dear Reviewer, thank you very much for you positive evaluation of our manuscript.

 Full names for ESR and CRP were added to the abstract.  Significant p-values were higlighted.

Conclusions were updated according to your recommendations. The short flow-chart created.

Dear Reviewer!

I hope the manuscript became better after your suggestions and recommendations.

On behalf of the Authors

Mikhail Kostik, MD, PhD, Professor

Reviewer 3 Report

Comments and Suggestions for Authors

The paper by Avrusin et al is an interesting paper, although came from a small sample.  The following changes should be done to improve the manuscript.

Major issues

1.      In the results the authors wrote paragraphs similar to this "Patients with HScore > 91 were more likely to have such symptoms as cervical lymphadenopathy (80.6% vs. 54.1%, p=0.0007)," although it is fine to put the percentages they should calculate Odds Ratio of prevalence with its 95% confidence interval. The calculations can be easily done with the OPENEPI program available for free on the internet. https://www.openepi.com/TwobyTwo/TwobyTwo.htm

2.      In Table 3 next to the estimate of sensitivity and specificity the 95% confidence interval should be included. This can also be easily done with the openepi program mentioned above, at the following link.  https://www.openepi.com/DiagnosticTest/DiagnosticTest.htm

Minor issues

In table 2, don't write p=0.000000 use instead p < 0. 000001

In line 177 don't write p=0.000000 use instead p < 0. 000001 (do the same in the whole paper)

In the footer of table 3, it should be indicated that Se and Sp correspond to sensitivity and Specificity.

In table 2 in the columns there is a number followed in parentheses by two numbers separated by a comma. Eg: Age, months 104 (68; 140) It should be explained in the title or in the foot of the table what it means (eg median and interquartile range, median and interquartile range, mean or 95% confidence interval, etc.) The table should be readable and understandable without reading the text of the article.

In the discussion it would be interesting for the authors to talk about future developments, e.g. using the same database calculating multivariate analysis, etc.

Author Response

Reviewer 3: The paper by Avrusin et al is an interesting paper, although came from a small sample.  The following changes should be done to improve the manuscript.

Dear Reviewer! Thank you so much for your positive evaluations of our manuscript. Our answers (A) on your queries (Q) are below and highlighted by color in the manuscript.

 Major issues

Q1.      In the results the authors wrote paragraphs similar to this "Patients with HScore > 91 were more likely to have such symptoms as cervical lymphadenopathy (80.6% vs. 54.1%, p=0.0007)," although it is fine to put the percentages they should calculate Odds Ratio of prevalence with its 95% confidence interval. The calculations can be easily done with the OPENEPI program available for free on the internet. https://www.openepi.com/TwobyTwo/TwobyTwo.htm

A1. Dear Reviewer! Thank you so much for your suggestion, improving the manuscript. OR with 95% CI were added to all mentioned parameters.

Q2.      In Table 3 next to the estimate of sensitivity and specificity the 95% confidence interval should be included. This can also be easily done with the openepi program mentioned above, at the following link.  https://www.openepi.com/DiagnosticTest/DiagnosticTest.htm

A2. Dear Reviewer! Thank you so much! The 95% CI included to the OR parameter and highlighted with green color.

Minor issues

Q3. In table 2, don't write p=0.000000 use instead p < 0. 000001

In line 177 don't write p=0.000000 use instead p < 0. 000001 (do the same in the whole paper)

A3. Dear Reviewer!  All the p=0.000000 were changed to p=0.000001 according to your recommendations.

Q4. In the footer of table 3, it should be indicated that Se and Sp correspond to sensitivity and Specificity.

A4. Dear Reviewer! The footer of table 3 was updated.

Q5. In table 2 in the columns there is a number followed in parentheses by two numbers separated by a comma. Eg: Age, months 104 (68; 140) It should be explained in the title or in the foot of the table what it means (eg median and interquartile range, median and interquartile range, mean or 95% confidence interval, etc.) The table should be readable and understandable without reading the text of the article.

A5. Dear Reviewer! Table 2 was also updated with the suggested information. The information about type of the quantitative data was added. The footnotes were modified: more abbreviations were added and they were placed in the alphabetical order.

Q6. In the discussion it would be interesting for the authors to talk about future developments, e.g. using the same database calculating multivariate analysis, etc.

A6.  Dear Reviewer! We did the multivariate analysis but the results are not so impressive due to small sample size and other factors mentioned in the Limitations subsection.  We have added some more information about the future perspectives.

Dear Reviewer, thank you very much for your thorough review and helpful recommendations. I hope the manuscript became better after your suggestions and valuable recommendations.

On behalf of the Authors

Mikhail Kostik, MD, PhD, Professor

Round 2

Reviewer 1 Report

Comments and Suggestions for Authors

The manuscript was partially improved. The manuscript is suitable for publication

Comments on the Quality of English Language

Minor grammatical mistakes were encountered

Author Response

Dear Reviewer!

Thank you for your positive review.

The English language was checked. The mistakes were fixed.

On behalf of the Authors

Ilia Avrusin, MD, PhD

Reviewer 3 Report

Comments and Suggestions for Authors

The paper has improved, and the authors have incorporated all comments. There is only a typo p=0.000000  should be p < 0.0000001, several times in the manuscript.

Author Response

Dear Reviewer!

Thank you for your positive review and useful recommendations.

Dear Reviewer!  All the p=0.000000 in the text were changed to p<0.000001 according to your suggestions.
These changes are highlighted with yellow color in the manuscript.

On behalf of the Authors

Ilia Avrusin, MD, PhD